# Drivers of Broad-Spectrum Antibiotic Overuse across Diverse Hospital Contexts—A Qualitative Study of Prescribers in the UK, Sri Lanka and South Africa

**DOI:** 10.3390/antibiotics10010094

**Published:** 2021-01-19

**Authors:** Carolyn Tarrant, Andrew M. Colman, David R. Jenkins, Edmund Chattoe-Brown, Nelun Perera, Shaheen Mehtar, W.M.I. Dilini Nakkawita, Michele Bolscher, Eva M. Krockow

**Affiliations:** 1Department of Health Sciences, University of Leicester, Leicester LE1 7RH, UK; 2Department of Neuroscience, Psychology and Behaviour, University of Leicester, Leicester LE1 7RH, UK; amc@le.ac.uk (A.M.C.); emk12@le.ac.uk (E.M.K.); 3Department of Clinical Microbiology, University Hospitals of Leicester NHS Trust, Leicester LE1 5WW, UK; david.jenkins@uhl-tr.nhs.uk (D.R.J.); Nelun.Perera@uhl-tr.nhs.uk (N.P.); 4School of Media, Communication and Sociology, University of Leicester, Leicester LE1 7JA, UK; ecb18@le.ac.uk; 5Tygerberg Academic Hospital and Faculty of Health Sciences, Stellenbosch University, Cape Town 7505, South Africa; shamehtar@gmail.com (S.M.); michelebolscher@gmail.com (M.B.); 6Department of Clinical Microbiology, Faculty of Medicine, General Sir John Kotelawala Defence University, Dehiwala-Mount Lavinia 10390, Sri Lanka; dilininak@gmail.com

**Keywords:** antimicrobial stewardship, antibacterial agents, hospitals, physicians, decision making, qualitative methods, UK, South Africa, Sri Lanka

## Abstract

Antimicrobial stewardship programs focus on reducing overuse of broad-spectrum antibiotics (BSAs), primarily through interventions to change prescribing behavior. This study aims to identify multi-level influences on BSA overuse across diverse high and low income, and public and private, healthcare contexts. Semi-structured interviews were conducted with 46 prescribers from hospitals in the UK, Sri Lanka, and South Africa, including public and private providers. Interviews explored decision making about prescribing BSAs, drivers of the use of BSAs, and benefits of BSAs to various stakeholders, and were analyzed using a constant comparative approach. Analysis identified drivers of BSA overuse at the individual, social and structural levels. Structural drivers of overuse varied significantly across contexts and included: system-level factors generating tensions with stewardship goals; limited material resources within hospitals; and patient poverty, lack of infrastructure and resources in local communities. Antimicrobial stewardship needs to encompass efforts to reduce the reliance on BSAs as a solution to context-specific structural conditions.

## 1. Introduction

Overuse of antibiotics is a widespread problem, contributing to the growing global threat of antimicrobial resistance (AMR). Research indicates that at least 30% of antibiotics prescribed in hospital settings may be inappropriate [1,2]. Inappropriate prescribing includes subtherapeutic doses, excessive treatment duration [3,4] and antibiotics that are medically not indicated (e.g., in case of viral illnesses) [5]. Other examples are unnecessary prescribing of antibiotics in situations where infections may clear without drug treatment, and the excessive use of broad-spectrum antibiotics (BSAs), which are effective against a wider range of pathogens than narrow-spectrum antibiotics (NSAs) at the cost of being stronger drivers of AMR [6].

Avoiding unnecessary use of BSAs and swapping to NSAs where possible are key targets of antimicrobial stewardship programs. National guidelines for BSA use have been widely developed and implemented, but guideline adherence is often suboptimal [7]. Antimicrobial stewardship programs aim to improve adherence to guidelines through the implementation of approaches such as the UK’s ‘Start Smart—Then Focus’ toolkit [8], and through interventions to promote behavior change including clinician education, audit and monitoring of BSA use, and restrictions and controls on BSA prescribing [9].

The focus of stewardship efforts on individual behavior has come under criticism in recent years, with the argument that behavior cannot be isolated from context [10]. The importance of considering local culture, institutional context and wider community setting in relation to antimicrobial stewardship efforts is underlined by evidence that the drivers of antibiotic overuse vary between settings and across countries [11,12,13,14]; the factors that impact on effective implementation of antimicrobial stewardship programs also vary across countries [15]. Broom and colleagues [14], based on a large qualitative study across five Australian hospitals, describe the personal, interpersonal and institutional drivers of overuse, including inter-professional dynamics, features of patient populations, and resource issues. They conclude that stewardship efforts need to address these broader social and structural contributing factors to overuse, and take local context into account.

Looking at the problem at the global level, others have argued for the need to fore-front structural and infrastructure constraints as drivers of antibiotic overuse. Research primarily conducted in lower-income countries shows that the reasons for antibiotic overuse include issues outside of the individual prescriber’s control such as accessibility of medication and the impact of poor local sanitation and hygiene on the patient population [16]. Willis et al. [17], based on research in East Africa, describe how antibiotics can be seen as having become a quick fix for structural problems. This includes the use of antibiotics as a solution to the problems arising from poor sanitation in communities and unhygienic healthcare facilities, and as a fix for the limitations of local healthcare systems.

This study aims to map the multi-level influences on BSA overuse across an international sample of high and low income, and public and private, hospital settings. We focus on exploring the structural factors that drive reliance on BSAs and how these differ across diverse contexts.

## 2. Materials and Methods

### 2.1. Design

Semi-structured interviews were conducted with hospital prescribers across seven hospitals in the UK, South Africa and Sri Lanka. Interviews were conducted by a total of three researchers, with one member of local research staff in each country conducting all of the interviews in that locality. Interviewers were trained in qualitative interviewing, and used a common topic guide (see Appendix B). The topic guide explored antibiotic prescribing decisions, the choice of BSA versus NSA, and perceptions of AMR. BSAs were defined as antibiotics with activity against a wide range of pathogens. We asked prescribers about their decision making related to BSAs, drivers for the use of BSAs, and the benefits of BSAs to various stakeholders. Written consent was obtained prior to interview. Interviews were conducted between 2017 and 2018, were conducted in English, and ranged in length between 20 and 80 min each. Interviews were audio recorded, transcribed verbatim, and anonymized prior to analysis. Debriefs about the research findings were provided by the research team to all participating hospitals.

### 2.2. Settings

Sri Lanka is a lower middle-income country with a public healthcare system along with widespread private healthcare provision; in rural areas, the population is relatively dispersed and resources limited. South Africa is a middle-income country with a public healthcare system along with private providers; health inequalities are very significant and the public healthcare system faces a high burden of demand from infectious diseases. The UK is a high-income country, where the majority of healthcare use is publicly funded [18]. In Sri Lanka, participants were recruited from three different hospital sites. Two of these, one private hospital and one public hospital, were located in a major city in Sri Lanka. The third hospital was publicly funded and located in a relatively remote area. In South Africa, participants were recruited from two different hospitals located in the urban area surrounding a major city. One hospital was publicly funded, the other one belonged to a chain of private hospitals. In the UK, participants were recruited from two NHS (publicly-funded) hospital sites; one hospital was a large city-based teaching hospital, and the second a smaller hospital in a suburban setting. As the majority of medical care in the UK is provided through the public sector, two public hospitals were selected in the UK.

### 2.3. Participants

Participants were sampled purposively across the three participating countries. We sampled hospital prescribers with experience of working with acute medical patients and in surgery, including staff with a range of roles, seniority, and expertise. Potential participants were identified and contacted by the local study lead in each hospital, and participants were also asked to identify colleagues to be invited to participate.

### 2.4. Analysis

All interview recordings were transcribed and analyzed by the UK-based research team. A constant comparative approach was used [19] in order to enable interpretative analysis, grounded in the data. This involved an ongoing process of making comparisons between different data extracts, and between emerging concepts. Starting with open, descriptive coding of a selection of transcripts, an initial coding framework was created using NVivo 11 Software [20]. This was followed by an iterative process of coding framework refinement and further coding, while taking into account existing literature and theoretical concepts. EK and CT conducted initial descriptive coding to inform the development of the coding framework; EK completed coding of the full data set. Narrative summaries were used to synthesize, compare, and interpret themes in the data. Regular team meetings were held to discuss the data, update coding categories to reflect emergent themes, and reflect on and develop interpretations.

## 3. Results

Interviews were conducted with 46 prescribers. We interviewed 18 participants in Sri Lanka (6 from each of the three participating hospitals), 13 participants in South Africa (7 from the private hospital and 6 from the public hospital), and 15 participants in the UK (8 in one site, 7 in the other) (Table 1).

We identified factors that impacted on BSA overuse across the participating hospitals. These included factors at the individual, social and structural levels. Quotes illustrating the factors are available in Appendix A. Table 2 maps out the extent to which each factor was described by participants as influencing BSA overuse in the setting in which they worked, based on qualitative assessment of participants’ accounts to identify strength and direction of influence. Social and structural drivers of overuse varied significantly across settings; our analysis focused on characterizing the key structural drivers across the different settings, and exploring how these factors resulted in reliance on BSAs.

### 3.1. Individual Factors Impacting on BSA Overuse

Individual-level factors include the cognitions, perceptions, and characteristics that influence prescriber decision making about prescribing BSAs. Participants described how, particularly in the setting of acute medicine, they often had to make treatment decisions quickly under conditions of diagnostic uncertainty. Although they were aware of the problem of AMR, this was balanced against the needs of individual patients, and fear of consequences for themselves in the event that a patient’s condition deteriorated. Prescribers characterized BSAs as ‘powerful’ and fast acting, and described how the broad range of action of BSAs increased the chances of effective treatment in the first instance. Universally across settings, participants described how using BSAs provided a solution to diagnostic uncertainty, reassured them that their patients would be safe, and reduced their own risk of censure from colleagues or of litigation.


*There’s definitely an advantage, in terms of potentially covering organisms that you were not necessarily expecting initially. (SA008, public hospital)*



*So broad spectrum I am [using] at the start is like a miracle, the patient gets better and it is like a miracle drug. (SL008, private hospital)*


These individual-level factors were evident in prescriber accounts across all settings (Table 2), although there were some differences in the impact of these factors across settings. Participants in South Africa, and to a lesser extent, Sri Lanka, described how resistant infections were increasingly part of their day to day practice, meaning that concerns about AMR were more salient. Participants working in the UK were less likely to describe AMR as a problem in their day to day practice. Training and education in antibiotic use and stewardship was seen as having an impact on improving BSA use, and correspondingly, a lack of training was seen as contributing to suboptimal BSA use. The extent of training varied across settings, with participants in Sri Lanka being least likely to describe post-qualification training related to antibiotic use or stewardship. Engagement with guidelines also varied: doctors in the public hospital in Sri Lanka were all highly engaged with the use of national guidelines; guideline use was described as relatively good in UK public hospitals, and doctors in hospitals in South Africa used guidelines although a variety of different guidelines were in use. Lack of engagement with guidelines was seen as problematic in the private hospital in Sri Lanka.

### 3.2. Social Factors Impacting on BSA Overuse

Social norms around antibiotic use and the culture within the hospital were seen as influential (Table 2): Sri Lankan participants working in the private hospital, and to a lesser extent, the public hospital described norms of widespread reliance on BSAs; in contrast, in the private hospital in South Africa, the hospital’s intensive stewardship program was seen to be effectively shifting local norms around antibiotic use. In the private hospitals in both countries, doctors’ clinical autonomy was a significant barrier to efforts to reduce BSA overuse. Participants from both private hospitals argued that doctors working in these settings were resistant to being challenged about their practice, but participants from the South African private hospital felt that, in some cases, the collegial nature of their relationships with colleagues could enable them to challenge others and discuss antibiotic treatment in a constructive way.


*Majority of people if we try to give feedback [on their prescribing] they would feel offended, people would feel offended and that becomes a territorial thing. (SL007, private hospital)*



*P7, MC: Amongst ourselves, we often talk about it, you know, if, if I see a prescription for [BSA], I’m like, Why, why did you go there? And then, they will say, Well, this patient’s had had this, they’ve got a recurrent infection, they’ve got sputums or whatever results, and this is now why they’ve gone to the next level. (SA007, private)*


Participants from all settings described how their prescribing practice was influenced by their colleagues, senior colleagues in particular; participants also reflected on the difficulties of challenging others’ practice. Hierarchies and inter-professional relationships could support good BSA use or act as a barrier to efforts to improve: junior doctors in the public hospitals described how seeking advice from seniors or expert colleagues could help in making appropriate antibiotic choices, but also reflected on the difficulty of challenging or acting out of line with their consultant’s preferences.


*I mean here, which is very consultant led, you give the antibiotic that the consultant tells you to. (UK009, public hospital)*


### 3.3. BSA Overuse as a Response to Structural Problems

In our analysis, we identified structural drivers of BSA overuse: these varied significantly across settings (Table 2). Reliance on BSAs was driven by: structural factors that created tensions with stewardship goals or perverse incentives for BSA use; limited material resources within hospitals; and poverty, lack of infrastructure, and lack of resources in local communities.

#### 3.3.1. Structural Drivers Creating Tensions with Stewardship Goals

Participants gave accounts of how their reliance on BSAs was driven by specific risks and vulnerabilities created by structural drivers in particular settings. In the UK, doctors felt especially vulnerable to the risks of missing sepsis due to the high national profile of this condition, the implementation of organizational interventions targeting sepsis, and the perception that missing a case of sepsis would have significant repercussions for the doctor concerned. BSAs, as universal and effective drugs, provided protection against this risk.


*Given that there’s a, it feels like there’s a pressure, whether it be national or local, for avoidable sepsis deaths, we tend to give [broad spectrum] antibiotics more often now, even there’s a bit of a grey area, because it’s deemed safer, perhaps on an individual level and a medical legal issue rather than for that patient, to give antibiotics. (UK006, public hospital)*


In private hospitals in both South Africa and Sri Lanka, the funding structure and operating models in private practice were in tension with stewardship goals, creating incentives for doctors to prescribe BSAs. Doctors were in competition with each other for patients: if they lost or were unable to attract patients, their career and earnings could be under threat. Using BSAs meant doctors could reliably achieve quick and effective results for patients, and also satisfy patient demand for ‘powerful’ treatment. BSAs were seen by participants as having currency for attracting and retaining patients: securing the reputation and income of both the individual prescriber and of their employing organization. The private hospital in South Africa had a well-developed stewardship program, but participants still felt vulnerable to patient dissatisfaction and potential loss of clients when they avoided the quick wins offered by broad-spectrum antibiotics.


*So the patient goes to this clinician and they get better faster [with broad spectrum antibiotics]. And the same patient after for some other disease is going to other clinician, he is giving a narrow spectrum antibiotics for a week or for 14 days, the patient got vexed: “why I should take for 14 days when [first consultant] is giving for [fewer]?” So in this kind of competition […] the more patient is going towards the person who is relieving them fast. (SL008, private hospital)*



*Well obviously the patients, the patient would want you to give them the best antibiotic that would make them feel better. Sometimes that’s not always appropriate. You know, sometimes you need to start with an antibiotic which is appropriate, and if it doesn’t work then upscale to a broader-spectrum antibiotic. And sometimes the patient might not always understand that. (SA002, private hospital)*


#### 3.3.2. Limited Hospital Resources

BSA overuse in public hospitals could arise as a result of limited material resources and institutional pressures. This theme was especially prominent in the accounts of participants from the lowest-resource settings. Participants from public hospitals in Sri Lanka described the challenges of working in an overcrowded hospital, where the risk of healthcare-associated infections was high due to lack of space and the poor conditions of the physical environment, and where the numbers of patients constantly exceeded capacity resulting in constant pressure to try to discharge patients. BSAs, as universal and quick-acting drugs, enabled doctors to act under these challenging circumstances, providing safer care and achieving quicker turnover to manage the high number of patients.


*We have to treat the patient as early as possible and to discharge the patient rather than like keeping them for a longer time in the hospital. […] It is very difficult to manage because we have a heavy patient turnover […] and also when patients they might give infection to another one and we have to prevent the spread of infection as well. So that’s, and so we have to like decide on, think of, broad spectrum. (SL015, public hospital)*


Limited microbiology facilities in some lower-resource settings could leave doctors dependent on BSAs. In the public hospital in Sri Lanka, participants emphasized that limited microbiology lab facilities meant they were commonly working with uncertainty (Table 2). BSAs could be used to provide effective treatment without the need for further testing.


*There are some issues with our [laboratory] set up as well, […] sometimes that there are restrictions on maybe the culture bottles available, the resources available. [When there are problems] getting some isolated so then we can’t go for the targeted one obviously. (SL003, public hospital)*


Although the problem of limited material resources within hospitals was more of an issue for prescribers in the very low-resource settings of public hospitals in Sri Lanka, the problem of limited resources was also described in the higher-resource settings of the public hospitals in South Africa and the UK. In these settings, participants described limited staff numbers, lack of continuity of care, and pressures to move patients through the system. In the context of a system that was stretched, BSAs provided a quick solution: they could be started and continued without incurring a delay, without requiring additional procedures and without requiring significant investment of senior staff time.


*You don’t need to contact people out of hours, you don’t need to get microbiology involved […] you just can prescribe it without having to go through hoops to get the right antibiotic. (UK006, public hospital)*


Another issue raised by doctors, particularly in higher-resource settings, related to a lack of alignment between acute hospital systems that had been designed to address presenting problems quickly and at high volume, and the complexities of patients with multi-morbidities. Doctors could resort to using BSAs as a simple action to deal with complex patients presenting in acute care, when there was a lack of clarity about underlying causes of symptoms coupled with a pressure to act.


*I would say this is increasingly the norm in acute hospital care, sadly. […] A system which has been designed to fit one patient with one problem, coming up against a population with a multiplicity of problems and complexity, which means the system is not responding well to the patients’ complexity and kind of resorts to a learned pattern of behaviour. So “oh, there’s something wrong with this patient, let’s give them some antibiotics and hope it gets better.” (UK002, public hospital)*


#### 3.3.3. Lack of Infrastructure and Resources in the Community

Finally, participants’ accounts illustrated the impact of patient poverty, lack of infrastructure, and lack of resources in local communities on BSA overuse: this theme was most highly prominent in accounts of providers working in lower-resource settings. A very significant problem in the rural hospital in Sri Lanka, also noted in relation to the other public hospitals in Sri Lanka and South Africa, was the fact that many patients were poor and needed to work to sustain their families. This meant people often delayed seeking healthcare until they reached a very critical state of illness, and had often taken unspecified first line or even second line antibiotics in insufficient doses and durations before coming into hospital. The lack of high-quality healthcare in the community was seen as contributing to the problem. Participants described how this led to an unavoidable reliance on BSAs when patients arrived at hospital, as BSAs could be easily and quickly deployed against severe infection.


*The patients […], they are very critically ill patients, because here, being a district where the literacy level is very low, and the education level is very low of the patients, and the infrastructure is also very low, so they often they present at the hospital quite at a late stage. So even when they come with an infection, if we delay the antibiotics [to await test results] the prognosis is definitely going to be poor. (SL006, public hospital)*



*I think our population in state [public hospitals] is much sicker, and they come at the end of their illness. […] When they present they’re quite sick, so they’re ready to be a sepsis, so you really have to start off with a broad-spectrum. And they come from locum clinics, where initially [they got] antibiotics, there was no cultures beforehand, so you’re sort of shooting in the dark. (SA002, private hospital)*


In the public hospitals in South Africa and Sri Lanka, patient poverty also shaped treatment choice. Participants recognized that patients needed to get well and return to work quickly to generate income for their families. BSAs produced quick and effective results—making a short length of stay more likely. Doctors were also sensitive to the issue of cost to patients in the private hospitals, where patients commonly had limited funds or, in the case of the Sri Lankan private hospital, a cap on reimbursement from their insurance scheme. As BSAs could be prescribed without the additional cost of running laboratory tests, and were likely to produce a quick result, this made them a better alternative when patients had limited funds.


*Well, cost to the individual patient. […] Broad spectrums are probably [more] expensive, but, getting it right the first time, again, might be cheaper than repeated doses of [antibiotics]. (SA007, private hospital)*



*[Investigations are] very costly so that is why rather than doing the [microbiology] investigation most of the doctors are directly prescribing the antibiotics. (SL008, private hospital)*


## 4. Discussion

BSA overuse is driven by factors at the individual, social and structural levels across settings. Individual and social influences on antibiotic use are well documented, including the role of risk perception, social norms, relationships with peers, and team hierarchy [14,21,22,23,24,25,26]. Our findings provide insight into how individual and social influences can vary across settings. Notable differences between settings at the individual level included the salience of AMR in prescribers’ day to day practice—lower exposure to resistant infections in UK hospitals meant stewardship was perceived by some as less urgent—as well as prescribers’ knowledge and education about antibiotic use, and engagement with guidelines. Social influences were most powerful in private hospital settings, where social norms around prescribing and clinical autonomy were strong drivers of BSA overuse. Across the settings, context-specific structural conditions acted as drivers of BSA overuse; these included dynamics within healthcare systems or organizations that created perverse incentives for BSA overuse, and resource limitations within hospitals and in the wider community. Our findings describe how BSAs, with their characteristic features of wide coverage, effectiveness, and ease of deployment, become used as a simple solution to challenges arising from structural constraints and limitations. Reliance on BSAs as a solution to these problems creates value for clinicians, patients, healthcare organizations, and society [27], but is in conflict with stewardship goals of tackling AMR.

Stewardship efforts, to some degree, aim to counter the appealing properties of BSAs as a quick solution to complex problems, for example, by putting limitations and restrictions around their use. Antimicrobial stewardship programs also focus on technical interventions including guidelines and monitoring, and, increasingly, incorporate evidence-based approaches to promoting behavior change such as education and restrictions on prescribing. These interventions aim to ‘correct’ suboptimal prescribing behavior and encourage rational antibiotic use. Evidence suggests that behavioral interventions are effective in improving antibiotic use [9,28], and calls have been made for better use of behavior change theory in the design of stewardship programs [29]. Behavior change interventions are a valuable component of antimicrobial stewardship programs, but stewardship needs to go beyond approaches that focus on individual prescriber behavior. Depending on the context within which doctors are working, BSA overuse can in some instances be understood as a rational and justifiable action by the individual doctor [12]. Reducing BSA overuse may be difficult or impossible for doctors when they are reliant on BSAs to maintain their customer base, or to enable them to do their job and effectively treat their patients in the face of adverse conditions. Interventions targeting prescriber behavior change risk ‘individualizing’ the problem [10,30]—placing the responsibility for optimizing antibiotic use on the clinicians, whereas the drivers of reliance on BSAs are multi-factorial and include social and structural factors outside of the control of the individual.

Antibiotic decision making is a collective, multi-disciplinary activity [31,32], and approaches to reducing BSA overuse need to involve all stakeholders in the hospital in changing the systems, processes and behaviors that result in reliance on BSAs. This will not be achieved through a focus on doctors alone, but requires doctors to be involved as part of wider a team across the organization, working together towards shared objectives, with strong leadership. It also requires the development of organizational infrastructure for optimizing BSA use including supporting structures, audit and feedback, and incentives that align with stewardship goals. The design of antimicrobial stewardship interventions need to take into account the pre-existing individual, social and structural conditions unique to each setting.

We also argue that antimicrobial stewardship approaches need to more broadly encompass efforts to reduce the reliance on BSAs as a response to local structural conditions. Although structural conditions are recognized as consequential for antibiotic overuse [14,17,33], this remains problematic for antimicrobial stewardship, as these types of problem are complex and difficult to address. Recommendations have been made for comprehensive approaches to stewardship, particularly in LMIC settings, that include increased attention to structural issues such as sanitation, infection prevention policy and planning, improvements to medicines regulation, and investment in diagnostic facilities and healthcare facilities [25,34,35,36]. Attempting to resolve these types of drivers of antibiotic overuse is likely to require extensive investment as well as regulatory and policy intervention. Some of the structural drivers of antibiotic overuse in higher income settings also present complex challenges, including the market forces driving overuse in private hospital settings [37], and the tensions for prescribers generated by multiple and potentially conflicting improvement goals [38].

Although wholesale structural change may be challenging, there is value in considering whether other feasible, low-cost, and contextually-sensitive solutions might be possible to help reduce reliance on BSAs. This might entail considering where to direct limited resources to best effect in LMIC settings, for example, towards enhanced cleaning in hospital environments, or targeting the delivery of adequate microbiology laboratory services to the patient groups where this will have the highest impact. However, it can also prompt us to ask different questions depending on the underlying drivers of overuse in different settings, for example:How can we ensure that doctors feel safe and supported to reduce BSA use in the context of organizational priorities and national drivers around reducing mortality from infection?How might we enable doctors to attract patients and succeed in private practice through building a reputation as a responsible prescriber of antibiotics?What low-cost interventions would help reduce the risk of infection, encourage help seeking, and enable early and effective treatment in resource-poor communities?How can we design infection control interventions that are feasible in suboptimal hospital environments?How could pricing systems in private hospitals be redesigned to remove perverse incentives for using BSAs (for example, reducing or removing charges to patients for lab tests)?

A strength of our research is the inclusion of prescribers working across a diverse range of settings; this enabled us to identify the drivers of BSA overuse across very different healthcare contexts. Participants were recruited through local contacts; we ensured that we included participants with a range of roles and seniority within sites, but participants do not represent the full range of prescribers. Further, we did not interview staff in other roles with responsibilities for antibiotic stewardship, e.g., nursing staff, microbiologists, pharmacists, or hospital managers. We interviewed a small number of staff in each setting, and our findings do not provide a definitive or comprehensive map of the drivers in each hospital, instead they provide an overview of patterns within and across settings in terms of key drivers of BSA overuse.

## 5. Conclusions

In conclusion, our research highlights drivers of BSA overuse and how they vary across different types of healthcare setting in countries with different health systems and levels of resource. We identified social and structural drivers that varied significantly across settings. These included structurally embedded risks and perverse incentives, social norms, missing infrastructure, and patient poverty. Our findings add weight to the argument that efforts to optimize antibiotic overuse need to go beyond a focus on correcting prescribing behavior in line with stewardship goals—as reliance on BSAs can arise as a response to local social and structural conditions that constrain the possibilities for action. Antimicrobial stewardship efforts should recognize the need for collective and organization-wide collaboration, and include a focus on identifying alternative, contextually-sensitive, solutions to the structural issues that drive reliance on BSAs.

## Figures and Tables

**Table 1 antibiotics-10-00094-t001:** Participant characteristics.

Characteristics	Participants
Role	Senior doctor, 20
	Junior doctor, 25
	Nurse, 1
Gender	Female, 18
	Male, 28
Specialty	General medicine, 27
	Surgery, 5
	Infectious diseases, 4
	Emergency department, 3
	Geriatrics, 3
	Hematology 1
	Oncology, 1
	ICU, 1
	Endocrinology, 1

**Table 2 antibiotics-10-00094-t002:** Influencing factors mapped by setting.

Theme	Impact on BSA Overuse				
	UK Public	South Africa Public	Sri Lanka Public	South Africa Private	Sri Lanka Private
**Individual factors**					
Diagnostic uncertainty	-	-	- -	-	-
Prioritizing reduction in risk for individual patients	-	-	-	-	-
Fear of repercussions from not prescribing	-	-	-	-	-
Perceptions of BSAs as ‘powerful’ and effective	-	-	-	-	-
Concern about AMR as a pressing problem	+/-	+	+/-	+	+/-
Training/knowledge/experience in antibiotic use	+/-	+/-	-	+/-	-
Engagement with antibiotic guidelines and policies	+	+/-	+	+/-	-
**Social factors**					
Social norms/culture of antibiotic use			-	+/-	- -
Clinical autonomy				-	- -
Hierarchy and willingness to challenge colleagues	-	-	-	+	-
**Structural factors**					
Conflicting quality and safety agendas	- -				
Pressure/incentives to satisfy patient demand	-			-	- -
Hospital environment		-	- -		
Limited microbiology facilities		-	- -		
Level of strain on the system/need for efficiency	-	-	- -		
Availability and quality of antibiotics in hospital			-		
Patient poverty		-	- -	-	-
Delayed presentation		-	- -		
Poor/uncontrolled community healthcare resources		-	- -	-	-

- Moderate contributing factor to BSA overuse; - - major contributing factor to BSA overuse; + supporting factor for reducing BSA overuse. If no symbol is shown, the theme was not mentioned as a significant issue.

## Data Availability

The data presented in this study are available in the UK Data Service on registration, at doi:10.5255/UKDA-SN-853655, reference number 853655.

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
