# Peer review of "Drivers of Broad-Spectrum Antibiotic Overuse across Diverse Hospital Contexts—A Qualitative Study of Prescribers in the UK, Sri Lanka and South Africa"

_antibiotics, 2021, doi:10.3390/antibiotics10010094_

Round 1

Reviewer 1 Report

This study aims to explore how and why broad-spectrum antibiotics are being used as a solution to structural problems in three different countries. Overall, the manuscript has been well written and the topic is highly important. However, I have some comments and queries.  

Background

The background has been written well and covered relevant literatures

Methods

The authors say they analysed data using a “constant comparative approach”? This method needs a bit more description for clarity and rationale of why this method was chosen?

Who and how many research members were the interviewers? Was the same or different interviewer team take the interviews as participants were sampled from three different countries?

Was the Interview guide translated into different languages? If yes, how did authors make sure the consistency in the data and also the reflexivity?

How many authors coded the data is unclear? What percentage of data was coded by second or third coder if this was the case.

Results

Results have been described nicely but I wonder whether any divergent themes appeared when UK data was compared with other two countries? As structural problems are a bit different among countries, perceptions of prescribers in using BSAs could have been different. Comparative results can be described a bit more if possible to understand context specific intervention approaches as a solution to optimise BSAs in future.

Convergent and divergent themes among countries can be separately be presented if analysed.

Author Response

Reviewer 1

Background

The background has been written well and covered relevant literatures

Thank you for this feedback.

Methods

The authors say they analysed data using a “constant comparative approach”? This method needs a bit more description for clarity and rationale of why this method was chosen?

More description and rationale have been provided, in section 2.4

Methods

Who and how many research members were the interviewers? Was the same or different interviewer team take the interviews as participants were sampled from three different countries?

Interviews were conducted by a total of three researchers, with one member of local research staff in each country conducting all of the interviews in that locality. We have added this information in section 2.1

Methods

Was the Interview guide translated into different languages? If yes, how did authors make sure the consistency in the data and also the reflexivity?

Interviews were conducted in English, as all medical professionals in each setting were English speakers. We have clarified in section 2.1

Methods

How many authors coded the data is unclear? What percentage of data was coded by second or third coder if this was the case.

The project researcher EK and the project lead CT conducted initial descriptive coding to inform the development of the coding framework; EK completed coding of the full data set. We have clarified this in section 2.4

Results

Results have been described nicely but I wonder whether any divergent themes appeared when UK data was compared with other two countries? As structural problems are a bit different among countries, perceptions of prescribers in using BSAs could have been different. Comparative results can be described a bit more if possible to understand context specific intervention approaches as a solution to optimise BSAs in future.  Convergent and divergent themes among countries can be separately be presented if analysed

Thank you for this suggestion, we agree that the paper will be strengthened by the inclusion of comparative data across settings.  We have expanded the findings to include a broader overview of the drivers of BSA overuse across the settings, and have included, in table 2, a comparison of the impact of different drivers across the different settings.  In describing our findings we have attempted to draw out differences between settings, and have reflected on them in the discussion.

Reviewer 2 Report

The authors set out to address the important issue of overuse of broad spectrum antibiotics (BSA) by leading interviews with prescribers working in different countries and different settings.

The approach of interviewing the people 'on the front' promises better understanding of their viewpoints, of the drivers of certain prescribing behaviors and potential barriers to adopt behaviors more suitable for microbial stewardship.

The paper is well written and easy to read.

However, in my opinion, there is room for improvement.

Line 88-92: Assuming that this paragraph defines the objective of the study, I find this a bit leading and limiting: "we aim to identify how BSAs overuse occurs as a response to structural constraints and limitations".
With the questions you asked in the interview and the target participants, I would have expected more something like "we aim to identify from the prescriber perspective the main drivers for BSA use and the barriers to implementing Microbial stewardship in the prescribing reality in different settings and countries."

Line 97: please add, in which language these interviews were conducted.

Line 103-105: Did the interviewees have a chance to review the transcribed version?
Did the interviewees receive the questions beforehand to prepare for the interview?

Line 115-120: What was the response rate?
How many potential participants were contacted and how many accepted? Were there any drop-outs ?

Why was it only prescribers (doctors) and not the other important stakeholders such as hospital pharmacists, hospital leadership, etc, who should also play a major role in Microbial stewardship?

Line 142-143: what were the other interviewees who were not doctors? What specialties were represented? Produce overview table on 'demographics'

Results section (starting line 138): In the methods section, the authors describe how a coding framework was created using NVivo 11 Software. It would be useful to see a bit of the (quantitative) results of the analysis to better understand the main drivers, barriers, or supporting factors in the different settings from the prescriber perspective in more detail. While some differences across settings were described ad verbatim it would be useful to see comparative tables, which summarise the differences across settings (e.g. country vs country; private vs. public).

When looking at the questionnaire, it also seemed that the majority of the interview responses were not reported in this manuscript (which may be connected to the somewhat biased objective of the study). For example, it would be valuable to understand the training / educational aspects and knowledge about the BSA issues, as well as responses to Q5, Q6, Q7, Q8, Q9, Q10, Q11, Q12, Q13, Q14, Q15.

Line 306-309: This is for me a critical sentence in this manuscript. Unfortunately, by only looking at the prescriber perspective, the other factors can also not be elucidated in this study.

Line 343: It would good to see the diversity described in a table. It would also be good to have a bit more contextual information about the hospitals, beyond public/private and country, to understand what the situation is.

General: while it is important to understand why doctors / prescribers continue to over-prescribe BSA, this is only one part of the story. Somehow, the conclusion of the study sounds to me like "It's not the doctors fault. First you need to solve all other problems and then you can blame the doctors." However, the actual point is that all player need to come together and align to improve the situation. There is no doubt that BSAs are very powerful treatments and there are many incentives to use them. However, the unfortunate issue is that there negative consequences, which are not immediately visible at the time of prescribing. All stakeholders in the hospital have to come together and collaborate to develop measures to change the processes and behaviors. This cannot be achieved by the doctors alone, but they need to be part of the team with the same objectives. That needs a strong support from the leadership and the other functions in the hospital; it needs supporting structures, quality control systems, nudging, as well as incentives, which point in the same direction.

In summary, I feel, that the authors could make more out of the research they have done by aiming a less biased  outcome and by describing the full findings in more detail and drill into more depth of  the analysis.

Author Response

Reviewer 2

Line 88-92: Assuming that this paragraph defines the objective of the study, I find this a bit leading and limiting: "we aim to identify how BSAs overuse occurs as a response to structural constraints and limitations".

With the questions you asked in the interview and the target participants, I would have expected more something like "we aim to identify from the prescriber perspective the main drivers for BSA use and the barriers to implementing Microbial stewardship in the prescribing reality in different settings and countries."

Our analysis focused particularly on structural drivers of BSA overuse, but we appreciate that the paper would be strengthened by taking a broader perspective on drivers of BSA overuse.  We have broadened the aims of the paper as suggested, while maintaining the emphasis on structural factors. We have revised the aims to state: ‘this study aims to map the multi-level influences on BSA overuse across an international sample of high and low income, and public and private, hospital settings. We focus on exploring the structural factors that drive reliance on BSAs and how these differ across diverse contexts.’

Line 97: please add, in which language these interviews were conducted.

Interviews were conducted in English, as all medical professionals in each setting were English speakers. We have clarified in section 2.1

Line 103-105: Did the interviewees have a chance to review the transcribed version?

Did the interviewees receive the questions beforehand to prepare for the interview?

We chose not to return the transcripts to interviewees for checking. Also, we did not send interviewees the questions beforehand to prepare for the interview. 

Line 115-120: What was the response rate?

How many potential participants were contacted and how many accepted? Were there any drop-outs ?

We purposively sampled participants in each setting to gain views from a range of staff in terms of their role and seniority.  In each site we used a variety of approaches to identify participants. Local leads identified interested participants; researchers visited participating sites to meet staff and share information about the study e.g. at unit meetings; we also used a snowball sampling approach by which people who had taken part in interviews identified colleagues who would be willing to take part. As such we did not assess response rate.

Why was it only prescribers (doctors) and not the other important stakeholders such as hospital pharmacists, hospital leadership, etc, who should also play a major role in Microbial stewardship?

We recognise that staff in a variety of roles play a part in antimicrobial stewardship, but our focus was on prescriber views of factors that contributed to overuse of BSAs. We have noted this as a limitation in the discussion.

Line 142-143: what were the other interviewees who were not doctors? What specialties were represented? Produce overview table on 'demographics'

We have provided more detail about interviewees in table 1.

Results section (starting line 138): In the methods section, the authors describe how a coding framework was created using NVivo 11 Software. It would be useful to see a bit of the (quantitative) results of the analysis to better understand the main drivers, barriers, or supporting factors in the different settings from the prescriber perspective in more detail. While some differences across settings were described ad verbatim it would be useful to see comparative tables, which summarise the differences across settings (e.g. country vs country; private vs. public).

Thank you for this suggestion. We have included an overview of individual, social and structural drivers (supplementary material), and a comparison of the impact of different drivers across the different settings (table 2).

When looking at the questionnaire, it also seemed that the majority of the interview responses were not reported in this manuscript (which may be connected to the somewhat biased objective of the study). For example, it would be valuable to understand the training / educational aspects and knowledge about the BSA issues, as well as responses to Q5, Q6, Q7, Q8, Q9, Q10, Q11, Q12, Q13, Q14, Q15.

In response to this feedback, and to feedback from reviewer 1, we have broadened the focus of the paper to include drivers of BSA overuse across individual, social and structural levels. This includes consideration of education and training, and engagement with guidelines across settings.

Line 306-309: This is for me a critical sentence in this manuscript. Unfortunately, by only looking at the prescriber perspective, the other factors can also not be elucidated in this study.

We acknowledge this and have added another sentence to the discussion to emphasise this.

Line 343: It would good to see the diversity described in a table. It would also be good to have a bit more contextual information about the hospitals, beyond public/private and country, to understand what the situation is.

We have already provided some information about the hospitals in the methods setting, and would prefer not to provide more information about the settings as this may compromise anonymity. We have added some information about country-specific factors that impact on antibiotic use across the three settings, in section 2.2, and added a reference to our paper that discusses the context for stewardship within each of these three countries in more depth.

General: while it is important to understand why doctors / prescribers continue to over-prescribe BSA, this is only one part of the story. Somehow, the conclusion of the study sounds to me like "It's not the doctors fault. First you need to solve all other problems and then you can blame the doctors." However, the actual point is that all player need to come together and align to improve the situation. There is no doubt that BSAs are very powerful treatments and there are many incentives to use them. However, the unfortunate issue is that there negative consequences, which are not immediately visible at the time of prescribing. All stakeholders in the hospital have to come together and collaborate to develop measures to change the processes and behaviors. This cannot be achieved by the doctors alone, but they need to be part of the team with the same objectives. That needs a strong support from the leadership and the other functions in the hospital; it needs supporting structures, quality control systems, nudging, as well as incentives, which point in the same direction

We have reworded the discussion to address this point, including a short paragraph:

‘Antibiotic decision-making is a collective, multidisciplinary activity [31, 32], and approaches to reducing BSA overuse need to involve all stakeholders in the hospital in changing the systems, processes and behaviors that result in reliance on BSAs. This will not be achieved through a focus on doctors alone, but requires doctors to be involved as part of wider a team across the organization, working together towards shared objectives, with strong leadership. It also requires the development of organizational infrastructure for optimizing BSA use including supporting structures, audit and feedback, and incentives that align with stewardship goals. The design of antimicrobial stewardship interventions need to take into account the pre-existing individual, social and structural conditions unique to each setting..’

In summary, I feel, that the authors could make more out of the research they have done by aiming a less biased  outcome and by describing the full findings in more detail and drill into more depth of  the analysis.

We hope that the material we have added to the findings, and revisions to the paper, have enabled us to do justice to our findings.

Round 2

Reviewer 2 Report

I would like to thank the authors for the revision. All my concerns have been addressed sufficiently and I fully support the publication.

Two small points:

Line 87: I am not sure whether the last comma is correct.

Line 323: insert 'the' before hospital